# Spatial variability in tropospheric peroxyacetyl nitrate in the tropics from infrared satellite observations in 2005 and 2006

Vivienne H. Payne<sup>1</sup>, Emily V. Fischer<sup>2</sup>, John R. Worden<sup>1</sup>, Zhe Jiang<sup>3</sup>, Liye Zhu<sup>2</sup>, Thomas P. Kurosu<sup>1</sup>, Susan S. Kulawik<sup>4</sup>

1

<sup>1</sup>Jet Propulsion Laboratory, California Institute of Technology, Pasadena, California, 91109, USA
 <sup>2</sup>Colorado State University, Department of Atmospheric Science, Fort Collins, Colorado, USA
 <sup>3</sup>National Center for Atmospheric Research, Boulder, Colorado, USA
 <sup>4</sup>Bay Area Environmental Research Institute, Mountain View, California, USA

Correspondence to: Vivienne H. Payne (vivienne.h.payne@jpl.nasa.gov)

**Abstract.** Peroxyacetyl nitrate (PAN) plays a fundamental role in the global ozone budget and is the primary reservoir of tropospheric reactive nitrogen over much of the globe. However, large uncertainties exist in how surface emissions, transport and lightning affect the global distribution, particularly in the tropics. We present new satellite observations of free

- 5 tropospheric PAN in the tropics from the Aura Tropospheric Emission Spectrometer. This dataset allows us to test expected spatio-temporal distributions that have been predicted by models but previously not well observed. We compare here with the GEOS-Chem model with updates specifically for PAN. We observe an austral springtime maximum over the tropical Atlantic, a feature that model predictions attribute primarily to lightning. Over Northern Central Africa in December, observations show strong inter-annual variability, despite low variation in fire emissions, that we attribute to the combined
- 10 effects of changes in biogenic emissions and lightning. We observe small enhancements in free tropospheric PAN corresponding to the extreme burning event over Indonesia associated with the 2006 El Nino.

## **1** Introduction

Peroxyacetyl nitrate (PAN) provides a thermally unstable reservoir for nitrogen oxide radicals (NO<sub>x</sub>), facilitating their longrange transport at low temperatures and eventual release in warmer regions of the remote troposphere where they most

- efficiently contribute to ozone (O<sub>3</sub>) production [*Singh and Hanst*, 1981]. PAN chemistry effectively reduces O<sub>3</sub> production in NO<sub>x</sub> source regions and increases it in remote regions of the troposphere [*Wang et al.*, 1998, *Fischer et al.*, 2013]. PAN is thought to be the dominant species in the reactive nitrogen budget over much of the globe [*Roberts et al.*, 2007], but is a particularly difficult compound to simulate in models due to the complexity of PAN chemistry and uncertainties in precursor emissions. Comprehensive in situ measurements of PAN are limited for the troposphere, particularly in the tropics [*Maloney*
- et al., 2001; Singh et al., 1996].

Since PAN abundance can be highly variable in space and time, it is difficult to know if presently available but limited insitu measurements are broadly representative. Satellite measurements offer a new opportunity to place constraints on our understanding, providing global coverage over multiple years. Global measurements of PAN have previously been obtained via thermal-infrared measurements from the limb-viewing Atmospheric Chemistry Experiment Fourier Transform

- Spectrometer (ACE-FTS) and Michelson Interferometer for Passive Atmospheric Sounding (MIPAS) satellite sensors [*Tereszchuk et al.*, 2014; *Moore and Remedios*, 2010; *Wiegele et al.*, 2012; *Pope et al.*, 2016]. Limb sounding measurements of PAN for limited time periods, but at relatively high spatial density, have also been made from the Cryogenic Infrared Spectrometers and Telescopes for the Atmosphere (CRISTA), flown on the Space Shuttle for two separate missions in 1994 and 1997 [*Ungermann et al.*, 2016] These observations provide information in the uppermost troposphere and lower
- stratosphere. Observations in the nadir viewing geometry can provide sensitivity to PAN lower in the troposphere, where its variable stability makes its role in O<sub>3</sub> production more important to understand. PAN is formed rapidly in biomass burning plumes, and isolated cases of elevated PAN in biomass burning plumes in the troposphere have been observed from the MetOp Infrared Atmospheric Sounding Instruments (IASI) and Aura Tropospheric Emission Spectrometer (TES) sensors [*Clarisse et al.*, 2011; *Alvarado et al.*, 2011]. More recently, global measurements of tropospheric PAN from Aura-TES have
- been obtained and are described in *Payne et al.* [2014]. These TES PAN retrievals have so far been utilized in studies of the influence of fires in atmospheric composition in boreal spring [*Zhu et al.*, 2015], the role of PAN in seasonal transport of East Asian pollution [*Jiang et al.*, 2016] and the seasonality and inter-annual variability of PAN in the Eastern Pacific [*Zhu et al.*, 2016]. Here we present new observations of TES PAN in the tropics. We focus on 2005 and 2006 in austral spring (the season of peak biomass burning) and compare these observations with simulations from the GEOS-Chem global chemical
- transport model.

The tropical troposphere plays an important role in global oxidation capacity, and understanding the role of PAN chemistry

is necessary to understand the different contributions to the  $NO_x$  reservoir and the  $O_3$  enhancement in the tropical south Atlantic. PAN can be formed within fire plumes because  $NO_x$  is co-emitted with large quantities of short-lived non-methane volatile organic compounds (NMVOCs), and it can form when biogenic NMVOCs react with  $NO_x$  produced by lightning. Formation of PAN in the cold upper troposphere over this region acts to sequester  $NO_x$  and decrease  $O_3$  formation. The

- contribution of biomass burning to the NO<sub>x</sub> reservoir and the O<sub>3</sub> enhancement in the tropical south Atlantic remains a long-standing issue [*Anderson et al.*, 1993; *Gregory et al.*,1996; *Jacob et al.*, 1996; *Edwards et al.*, 2003; *Ziemke et al.*, 2009]. Models predict that lightning is the most important source of PAN in the atmosphere of the tropical south Atlantic (*Fischer et al.*, [2014] and references therein). However, this finding is particularly sensitive to the description of boundary layer chemistry, which remains very uncertain [*Hewitt et al.*, 2010]. Implementation of a state of the science isoprene scheme
- [Paulot et al., 2009a, 2009b] reduces the model sensitivity of upper tropospheric PAN over the tropical Atlantic to lightning by changing the fraction of isoprene oxidized outside the boundary layer [Fischer et al., 2014]. Elevated PAN mixing ratios (~500 pptv) were observed in the mid- to upper troposphere over the tropical south Atlantic during the October 1992 TRACE-A aircraft campaign, and an austral spring maximum in this region is predicted by state-of-the-science global chemical transport models [Fischer et al., 2014; Fadnavis et al., 2014]. Limb-viewing satellite observations have shown
- PAN mixing ratios of ~350 pptv at 260 hPa in this region in austral spring [Moore and Remedios, 2010; Glatthor et al., 2007]. PAN observations over multiple years, in conjunction with global chemical models, offer potential to shed light on the influence of fire emissions on the inter-annual variability of the tropical south Atlantic O<sub>3</sub> maximum.

Section 2 describes the characteristics of the TES PAN retrievals, while Section 3 provides background on the GEOS-Chem model simulations used in this work. Section 4 describes the features observed by TES in the tropics in austral spring of 2005 and 2006. Section 5 presents the relationships between PAN and carbon monoxide (CO) in different regions and

20 2005 and 2006. Section 5 presents the relationships between PAN and carbon monoxide (CO) in different regions and discusses model/measurement comparisons. Conclusions are presented in Section 6.

#### 2. TES PAN retrievals

The Tropospheric Emission Spectrometer (TES) has been flying on the Aura satellite since 2004. TES measures nadirviewing spectrally resolved thermal infrared radiances, providing information on numerous trace gases in the troposphere, including PAN. The TES PAN retrievals use an optimal estimation approach. An algorithm description is provided in *Payne et al.* [2014]. TES has been shown to be capable of observing PAN with sensitivity to elevated concentrations (greater than  $\sim 0.2-0.3$  ppby) in the free troposphere (between  $\sim 800$  mbar and the tropopause). Estimated single-observation errors are 30

- 50 %. The number of degrees of freedom for signal (DOFS), or independent pieces of information, in the TES PAN retrievals is less than 1.0, meaning that the retrievals are not sensitive to the vertical distribution of PAN in the atmosphere.
- As discussed in *Payne et al.* [2014], TES PAN retrievals are generally insensitive to near-surface variations of PAN and are sensitive primarily to variations in the free troposphere. TES PAN retrievals are being processed routinely for the whole TES dataset and are publicly available in the TES v7 Level 2 product. However, at the time of this work, the v7 product was not
  - 4

yet available. The TES PAN retrievals shown here were processed using a prototype algorithm for the areas and time periods of interest.

PAN retrievals are not attempted for all TES targets. As discussed in *Payne et al.* [2014], PAN retrievals are not attempted for cases where the water vapor or  $O_3$  from previous retrieval steps did not pass the master quality flags. PAN retrievals are

- also generally not attempted over sandy or rocky surfaces, such as desert or mountainous regions. The reason for this is the presence of a silicate feature in the surface emissivity spectra of those surfaces that coincides with the spectral position of the PAN absorption feature. While this is not an issue for the tropical data, we note that PAN retrievals over icy or snowy surfaces are subject to a high bias. Again, this is due to spectral features in the emissivity for these surfaces. Therefore, we recommend screening out data with surface temperature less than 270 K. *Jiang et al.* [2016] performed indirect comparisons
- of TES PAN with aircraft measurements from the Arctic Research of the Composition of the Troposphere from Aircraft and Satellites (ARCTAS) campaign, using the GEOS-Chem model as a transfer standard. Results of that study suggest a high bias in TES cases with surface temperatures below ~280 K, where surfaces are not ice- or snow-covered. One possible explanation is that surface temperature is a proxy for the representative temperature of the column, and that the low bias stems from a lack of information on temperature-dependence in the HITRAN 2008 spectroscopic cross-sections used in the
- TES retrievals. The HITRAN 2012 database includes low temperature cross-section information for PAN. This will be considered for future versions of the TES algorithm. Based on the *Jiang et al.* [2016] comparisons for cases with warmer surfaces/atmospheres, we do not expect strong biases for the tropical data shown here.

We define cases for which elevated PAN is detected with confidence as those that pass basic quality checks and where the DOFS of the retrieval is greater than 0.6. Note that the use of DOFS is not, in itself, a quality flag. Retrievals with DOFS <

0.6 may converge with a good quality of fit. However, in those cases the retrieved PAN would be strongly affected by the prior constraint chosen for the retrieval. Further justification of the choice of DOFS=0.6 as a threshold can be found in *Payne et al.* [2014].

#### 3. GEOS-Chem global chemical transport model

GEOS-Chem (www.geos-chem.org) is a global chemical transport model driven by GEOS assimilated meteorological data from the NASA Global Modeling and Assimilation Office (GMAO). GEOS-Chem includes a state-of-the science description of tropospheric oxidant chemistry. We used v9.01.01 with updates specifically for PAN as described in *Fischer et al.* [2014] (and references therein) to explore, analyse and explain the global TES PAN data. GEOS-Chem was driven by NASA GEOS-5 assimilated meteorological data with 0.5° × 0.67° horizontal resolution, 47 vertical levels, and 3 – 6 hour temporal resolution. We degraded the horizontal resolution to 2 ° × 2.5 °. The simulations for 2005 and 2006 were preceded by a 1-

30 year spin-up.

Briefly, the version described in *Fischer et al.* [2014] includes updated budgets of many NMVOC PAN precursors including acetone, ethane and propane, acetaldehyde and methylglyoxal. Terrestrial biogenic emissions of NMVOCs are calculated using the Model of Emissions of Gases and Aerosols from Nature (MEGAN v2.0) [*Guenther et al.*, 2006]. The model incorporates a new oxidation schemes for isoprene, and several additional NMVOC tracers (monoterpenes, ethanol, aromatics) that also serve as PAN precursors. Other relevant updates include the treatment of emissions from fires. In particular, the model includes biomass burning emissions of shorter lived NMVOCs (monoterpenes, aromatics), 40% of the

particular, the model includes biomass burning emissions of shorter lived NMVOCs (monoterpenes, aromatics), 40% of the biomass burning NO<sub>x</sub> is directly emitted as PAN [*Alvarado et al.*, 2010], and 35% of fire emissions are injected into the 10 model layers above the boundary layer [*val Martin et al.*, 2010].

### 10 4. Observations of PAN in the Tropics

5

Figure 1 shows Aura-TES PAN in the Tropics for October 2006. Figure 1(a) shows individual observations. Volume mixing ratios (VMRs) in Figure 1(a) represent an average between 800 hPa and the tropopause. Figure 1(b) shows the fraction of observations with elevated PAN. This fraction is the ratio of the number of TES targets for which elevated PAN is detected with confidence to the number of targets for which the PAN retrieval was attempted. Figure 1(b) was created by calculating

- the fraction of observations with elevated PAN in  $4^{\circ} \times 5^{\circ}$  boxes, then smoothing this field with a two-dimensional boxcar average with a width of two boxes. For October 2006, we see a high density of elevated PAN detections over the tropical south Atlantic and the surrounding landmasses and a high fraction of TES observations with elevated PAN over the tropical south Atlantic. High PAN over the tropical south Atlantic in austral spring is one of the major features in the global PAN distribution predicted by the GEOS-Chem [*Fischer et al.*, 2014]. High PAN values over the tropical south Atlantic in the
- uppermost troposphere (~8-16 km) have previously been observed from MIPAS [*Moore et al.*, 2010; *Glatthor et al.*, 2007; *Pope et al.*, 2016]. The TES observations presented here provide information on the temporal evolution in austral spring 2005 and 2006. We express the gridded results in "fraction of observations with elevated PAN", rather than averaged PAN retrieval values, because the Aura-TES PAN retrievals are only possible for elevated PAN values. The precise details of the detection threshold depend on a number of factors, including cloud optical depth, the vertical distribution of PAN in the
- atmosphere and the details of the surface and atmospheric temperatures. Since water is a significant interferent in the spectral region used for the PAN retrieval, there is also some dependence of the detection threshold on the details of the water vapour profile. The vertical sensitivity of the TES PAN retrievals also varies with these factors, although in general the retrievals have highest sensitivity to variations in PAN in the free troposphere. Based on simulations over a range of conditions, *Payne et al.* [2014] specify an approximate detection threshold of 0.2 ppbv. Therefore, the interpretation of any averaged values
- would be complicated by the fact that we do not have information on the values at the low end of the true distribution.

Figure 2 shows histograms of free-tropospheric average PAN values in different regions of the tropics for September through December 2005 and 2006. Boxes showing the geographical extent of each of these regions are shown in Figure 1(a).

Histograms are calculated on a logarithmic scale, in order to better allow examination of differences in the 0.1 to 0.3 ppbv range. Histograms are normalized by the total number of TES observations in each region. Also shown for each region is the total of the histogram for each month and region. This total equates to the fraction of TES observations where elevated PAN was observed. It is clear from Figure 2 that there is considerable variation in free tropospheric PAN between regions and

- 5 from one month to the next. In general, higher PAN values and higher fractions are observed for the months where peak biomass burning occurs in those regions. For the Amazon and southern Africa, peak burning occurs in September and October. For northern Africa, peak burning occurs in December. For Indonesia, peak burning usually occurs between September and November. For 2006, the Indonesian fires began in October and persisted through November [*Logan et al.*, 2008]
- In all TES retrievals, an effective cloud optical depth is retrieved in order to mitigate the impact of clouds [Kulawik et al., 2006]. The impact of clouds is to reduce the sensitivity of the measured radiance to the target trace gas concentrations. This is accounted for in the averaging kernels and therefore is reflected in the DOFS for the retrieval. As discussed in *Payne et al.* [2014], clouds with optical depth greater than ~0.5 have the potential to obscure PAN signals that are comparable in magnitude to the instrument noise. We would therefore expect that the fractions shown in Figure 1(b) and Figure 2 would be, if anything, an underestimate of the true incidence of elevated PAN in the atmosphere.

For all the months shown here, TES made global survey measurements throughout the tropics, and the sampling is vastly more spatially uniform than could be obtained by any kind of in situ sampling strategy. However, the number of TES measurements in the tropics does vary somewhat between the two years shown here, with a greater number of measurements taken in 2006 than 2005. The number of measurements in any given region does also vary from one month to the next,

- 20 depending on the details of instrument operation. In both 2005 and 2006 there were significantly fewer global survey measurements taken in September than in other months. For example, in the latitude band between 30S and 10N, there were 6665, 10495, 10738 and 9486 TES measurements taken in September, October, November and December 2006 respectively. In September 2005, the measurements are distributed earlier in the month, while in September 2006, the measurements are generally later in the month. It is possible that this difference in temporal sampling could account for the observed year-to-
- 25 year differences in the fraction of elevated PAN over the Amazon region (South America) in September (see Figure 2). For October, November and December, the TES measurements are spread more evenly throughout each month in both years.

In terms of year-to-year differences, strong differences are observed for Northern Central Africa in December. December 2005 shows elevated PAN detected with confidence in 45 % of TES observations compared to 30 % in December 2006. GEOS-Chem simulations indicate that PAN concentrations in this region are strongly influenced by biomass burning (see

- 30 Supplement). The TES CO in this region does not show marked differences between 2005 and 2006 [Logan et al., 2008]. We infer from this that the observed year-to-year difference in PAN is not dominated by differences in biomass burning. MEGAN (via GEOS-Chem see Figure 3) does show higher monthly mean isoprene emissions over this region in
  - 7

December 2005 versus 2006. The difference in isoprene emissions at specific locations in the orange box in Figure 1a range from 10 - 50%. The total isoprene emissions for the region were ~13% higher in December 2005 than in 2006. Since biogenic emissions in the presence of lightning lead to PAN formation, stronger biogenic emissions in 2005 could contribute to higher PAN values. *Logan et al.* [2008] also note that there was more lightning over much of Africa (including the region

5

considered here) in 2005 compared to 2006. GEOS-Chem simulations showing the sensitivity of PAN to lightning for December 2005 and 2006 are shown in the Supplement. More lightning  $NO_x$  in December 2005 than 2006 would also lead to enhanced PAN.

Vertical transport is also a consideration. If the surface emissions were the same, we would expect that stronger convection in a given year would enhance the impact of surface emissions on mid-upper tropospheric PAN. It would not only enable
more efficient lofting of fire smoke, but would also allow the same quantity of biogenic NMVOC emissions and/or secondary products to contribute more efficiently to PAN formation aloft for a given amount of lightning NO<sub>x</sub>. Either way, stronger convection in a given year would increase the contribution of surface emissions to PAN in the mid-troposphere, where it can be observed with the nadir-viewing thermal infrared satellite measurements. Previous studies (e.g. *Nassar et al.*, [2009]) have pointed to the difference in convection over Northern Central Africa between these two years, and subsequent

- differences in O<sub>3</sub>. Nassar *et al.* note that convection was stronger in December 2006 than December 2005 in this region. This would act in the opposite direction to the observed year-to-year PAN differences. We did GEOS-Chem simulations without convection and found that the amount of PAN above Northern Central Africa is very sensitive to the presence of convection. Transport and scavenging in convective updrafts is coupled in GEOS-Chem [*Liu et al.*, 2001]. Turning off the convection operator effectively suppresses both convective transport and scavenging in updrafts. Other related process,
- e.g., lightning, NO<sub>x</sub> emissions, in-cloud oxidation all remain. Figure 4 shows maps of the difference in PAN between GEOS-Chem simulations with and without convection. In a global context, Northern Central Africa is one of the most sensitive regions. The enhanced convection in November and December 2006 would have acted to increase mid-tropospheric PAN in this region more strongly in 2006 than in 2005. However, the PAN over this region is in fact higher in 2005 than in 2006... Therefore, we conclude that the December year-to-year PAN difference in this region is most likely associated with changes
- in biogenic emissions and lightning.

A noticeable difference is also observed for Indonesia in October/November, with distinctly higher PAN in 2006 compared to 2005. *Logan et al.* [2008] have previously discussed extreme CO enhancements in October 2006, associated with the strong 2006 El Nino. During an El Nino event, the normally warm waters and associated convection over the western Pacific and maritime continent move towards the eastern Pacific, resulting in changes in the large-scale circulation. El Nino events are associated with decrease in convection and in precipitation over the maritime continent. The 2006 El Nino was associated with a severe drought in Indonesia, leading to intense fires in this region. The strong enhancements in the CO

8

discussed by Logan et al. [2008] extended into the upper troposphere and lower stratosphere, as seen by the Aura Microwave

Limb Sounder (e.g. *Zhang et al.*, [2011]). Given the large CO emissions from these fires, and the evidence of transport of the CO to upper altitudes, we might have expected to also observe elevated PAN over Indonesia in October/November 2006. The relationship between PAN and CO for this region/month is further discussed in Section 5. Logan et al. [2008] also highlight that there was more lightning (and therefore higher lightning NO<sub>x</sub> emissions) in November and December 2006

than in 2005 by a factor of 2-3, although there was less lightning in October 2006 compared to 2005. Differences in

convection and lightning between 2005 and 2006 in this region are further discussed in Nassar et al. [2009].

5

#### 5. PAN/CO enhancement and comparisons with GEOS-Chem

In order to further explore the role of biomass burning on the observed PAN, we use coincident TES measurements of carbon monoxide (CO). Figures 5(a) and (c) show scatter plots of TES-retrieved CO versus PAN, for selected regions for October 2005 and 2006. In general, elevated CO in tropical regions can be interpreted as an indication of strong fire 10 emissions. Variability in enhancements in PAN relative to CO ( $\Delta PAN/\Delta CO$ ) in fire plumes is driven by the efficiency of PAN formation, mixing [Yokelson et al. 2013] and transport. For example, in an evaluation of models at high latitudes, Arnold et al. [2015] note that model enhancement ratios show distinct groupings according to the meteorological data used to drive the models, which they show is likely linked to differences in vertical transport. Alvarado et al. [2010] discuss 15 aircraft in situ measurements of  $\Delta PAN/\Delta CO$  in fire plumes sampled during the ARCTAS aircraft campaign. In these highlatitude measurements, Alvarado et al. [2010] report lower values of  $\Delta PAN/\Delta CO$  for samples close to the fires, with higher values for samples in the plumes downwind, suggesting PAN formation within the plume. Gray dotted lines show maximum and minimum values of  $\Delta PAN/\Delta CO$  enhancements in the aircraft measurements of boreal fire plumes reported in *Alvarado* et al. [2010], and assuming a background mid-tropospheric CO value of 50 ppbv. These lines are shown here primarily to demonstrate the large range of values in aircraft observations of boreal plumes, not for the purposes of quantitative 20

comparison with these tropical satellite observations. The TES measurements shown in Figure 5 have not been specifically screened to establish fire influence, nor have attempts been made here to categorize the satellite measurements according to distance from fires.

We also compare the TES-retrieved PAN-CO relationships with those from GEOS-Chem. The PAN-CO relationship from GEOS-Chem for October 2005 and 2006 are shown in Figure 5 (b) and (d). When comparing GEOS-Chem modelled PAN with Aura-TES observations, we sampled the model fields at the measurement locations and times. The TES averaging kernels and *a priori* were applied to the GEOS-Chem profiles in order to account for the sensitivity of the TES measurements. Both the TES measurements and the GEOS-Chem model show a range of  $\Delta$ PAN/ $\Delta$ CO ratios. A considerable number of points show  $\Delta$ PAN/ $\Delta$ CO enhancements higher than that previously observed in biomass burning plumes in other

30 regions [*Alvarado et al.* 2010]. We hypothesize that high  $\Delta PAN/\Delta CO$  enhancements could also conceivably be associated with a strong influence of lightning. Unlike during fires, lightning NO<sub>x</sub> is emitted without CO.

The absolute values of PAN are distinctly higher in the measurements than the model. There are a number of possible reasons why the model might predict lower values than observed. One possible reason is a high bias in the observations. Pope et al. [2016], in a comparison between MIPAS PAN results from two different retrieval algorithms, found significant differences in the tropical PAN fields between the two sets of results and pointed to potential reasons for differences that

include differences in the way that PAN cross-section data are interpolated within the forward model used in the retrieval 5 algorithm. They concluded that the MIPAS satellite observations are able to detect realistic spatial variations in PAN, but that further work is needed to evaluate the satellite retrievals in an absolute sense. We acknowledge that this type of further work is also desirable for evaluation of the results from the TES PAN algorithm. Alternatively, the global model, with its limited spatial resolution, may not be able to capture relatively small-scale plume enhancements that could be observed by

10

the satellite [Rastigejev et al., 2010]. It is also possible that the fire injection heights in the model are inaccurate. Other possibilities include underestimation of the  $NO_x$  to PAN conversion ratios in the model or underestimation of the  $NO_x$ emissions themselves, either from fires, lightning or both.

Although the absolute PAN values from TES are higher than those from GEOS-Chem, we note that both model and measurements show features that are qualitatively consistent in terms of the  $\Delta PAN/\Delta CO$  relationship. Both model and measurements show a distinctive signature associated with the October 2006 Indonesian fires, extremely elevated CO and 15 distinctly low  $\Delta PAN/\Delta CO$  enhancement ratios. The low  $\Delta PAN/\Delta CO$  could be due to two factors: 1) We expect a higher emission ratio of CO relative to NO<sub>x</sub> for peat burning compared to both tropical forests and crop residue [Akagi et al., 2011; Stockwell et al., 2014], and 2) these plumes were not directly injected into the free troposphere, promoting the decomposition of PAN [Tosca et al., 2011].

- We used GEOS-Chem to assess the sensitivity of the model to injection height. For October 2005 and 2006, runs were 20 performed both for the default case where 35 % of the of fire emissions are injected into the 10 model layers above the boundary layer and for the case where all fire emissions are injected directly into the planetary boundary layer (PBL). We found that at least over Indonesia, the modelled free tropospheric PAN is not strongly sensitive to the injection height. The difference between the PAN for two runs was 10% at most (see Figure 6). A possible reason for this is that the persistent convection in this region enables rapid lofting of PAN to the free troposphere, regardless of whether the fire injection heights 25
- are within the boundary layer or above it. The model sensitivity result suggests that the higher emission ratio of CO relative to NO<sub>x</sub> for peat burning compared to tropical forests/crop residue is the dominant reason for the low  $\Delta PAN/\Delta CO$  observed for the Indonesian fires.

When similar runs were performed for December in North Central Africa (not shown), the difference in free tropospheric PAN was up to 40 %, indicating that sensitivity to injection height is stronger in that region. The temperature in the lower 30 atmosphere may also factor into the difference in sensitivity to injection heights between different regions.

## 6. Conclusions

Our findings can be summarized as follows: We observe elevated free-tropospheric PAN over the Tropical South Atlantic in austral spring, for the two years investigated (2005 and 2006). This feature has been predicted by models, and previously

- observed in MIPAS satellite observations of the uppermost troposphere. The TES observations presented here provide confirmation that this feature is also observed in the nadir view. We see a strong enhancement in PAN over Northern Central Africa (5°S to 10°N) in December 2005 relative to December 2006. Since convection was stronger in December 2006 than December 2005 in this region, we hypothesize that the December year-to-year PAN difference in this region is most likely associated with changes in biogenic emissions and lightning. We observe small enhancements in free tropospheric PAN and
- high enhancements in CO in October/November 2006 compared to 2005, corresponding to the extreme burning event over Indonesia associated with the 2006 El Nino. Comparisons between the TES observations and the GEOS-Chem model show qualitative agreement in observed regional and year-to-year variations in ΔPAN/ΔCO enhancement ratios.

Knowledge of the PAN distribution is key to understanding the reactive nitrogen  $(NO_y)$  budget that controls the tropospheric  $O_3$ . These new nadir-viewing satellite observations of PAN, analyzed in conjunction with a global chemical transport model,

demonstrate the importance of emissions, chemistry and transport in understanding the large-scale distribution of PAN. TES PAN retrievals will be routinely processed for the entire TES data record, from 2004 to the present, in the TES version 7 data release. We suggest that nadir satellite observations of PAN will complement the existing limb satellite observations and will provide a powerful tool in understanding the reactive nitrogen budget and the global transport of pollution from polluting to receptor regions.

#### 20

## **Competing interests**

The authors declare that they have no conflict of interest.

#### Acknowledgements:

TES PAN and CO data are archived at the NASA Langley Research Center Atmospheric Science Data Center (https://eosweb.larc.nasa.gov/project/tes/tes\_table). The TES products can also be accessed via the NASA Reverb tool (http://reverb.echo.nasa.gov). TES monthly Lite files are also available via the Aura Validation Data Center (http://avdc.gsfc.gov). The PAN dataset used in this work, produced using a prototype algorithm developed prior to the TES V07 Level 2 release, may be obtained upon request from the corresponding author (vivienne.h.payne@jpl.nasa.gov). Part of this research was carried out at the Jet Propulsion Laboratory, California Institute of Technology, under a contract with the National Aeronautics and Space Administration. Reference herein to any specific commercial product, process or service by trade name, trademark, manufacturer or otherwise does not constitute or imply its endorsement by the United States Government or the Jet Propulsion Laboratory, California Institute of Technology. This work was supported by NASA Award Number NNX14AF14G.

5 Copyright 2016. All rights reserved.

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
