# Peer review of "Spatial variability in tropospheric peroxyacetyl nitrate in the tropics from infrared satellite observations in 2005 and 2006"

_Atmospheric Chemistry and Physics, 2016_

## Referee Comment (RC1) · Anonymous Referee #2 · 19 Jan 2017

The authors present results from a recently developed and novel peroxyacetyl nitrate dataset from the TES instrument on Aura. The paper builds on previous work to demonstrate the ability of TES to measure PAN in enhanced conditions and presents an analysis of spatial and temporal variability of elevated PAN values in the tropics for 2005 and 2006. The method for producing the PAN data is discussed in a previously accepted paper in AMT. Data presented for 2006 are from an El Nino year and implications for PAN during increased fire activity periods are discussed. The paper presents comparison to model expectations during the same period and data between the observations and model are fairly consistent, at least qualitatively. The PAN data presented is currently unique for nadir viewing instruments and are of great value for testing the ability

of Earth system models to predict tropospheric ozone, for example. I have a few general comments and once these issues are addressed I am happy for the paper to be published in ACP.

General comments:

1) I would like more expansion of the model/measurement comparison (section 5). It is presented that absolute values of PAN are higher in the measurements than the model. Why does the model predict lower values? Where the measurements can help inform the model about where the formation mechanisms are lacking? I understand it is a data driven paper, but it is essential to discuss how these new datasets may be used to improve model predictions.

2) It is clear that PAN can only be measured in "elevated" concentrations and so it is not possible to look at regional averages etc. It would be useful to re-iterate what the detection limit is for this retrieval, without having to read through the original data paper. Does this limit vary from region-to-region based on thermal contrast, for example?

Specific comments and technical corrections:

1) Page 5, line 4: change to "GEOS-Chem (www.geos-chem.org) is"

2) Page 6, line 32: please add in which months peak biomass burning occurs either here or earlier in the text. It seems a little vague at the moment as the reader may not know which months are inferred.

3) Page 8, line 7: muddled sentence. I do not know what the authors are trying to say in this line, please rewrite.

4) Page 8, line 8: muddled sentence starting "The year-to-year...."

5) Page 9, line 26: muddled sentence "of the of fire", please adjust to make sentence clear.

---

## Referee Comment (RC2) · Anonymous Referee #1 · 22 Jan 2017

The manuscript presents a distribution of PAN in November/December 2005 and 2006 from TES measurements. Observations of PAN are available from a few satellites, e.g., ACE-FTS, MIPAS and AURA-TES. These measurements are valuable in understanding role of PAN chemistry. This paper presents a springtime maximum in PAN over North Central Africa in December 2005 compared to 2006. It states that strong convection in 2006 might have contributed to the observed low concentrations of PAN. A Small enhancement of PAN and high in CO over Indonesia in December 2006 has been attributed to extreme biomass burning associated with El Nino. It hypothesized that $\Delta PAN/\Delta CO$ enhancement could be associated with a strong influence of lightning. This paper provides interesting results and is thus suitable for ACP. I suggest

incorporating the following points:

(1) I suggest performing additional simulations from GEOS-Chem model, with and without lightning parameterization. Also, a set of simulations with biomass burning emissions on and off. It will help to justify hypothesis of the strong influence of lightning and Biomass burning.

(2) El Nino events are associated with less convection (droughts) and lightning. An elaborated discussion and a supporting figure should be incorporated.

(3) Details of convection scheme switched on and off in GEOs-Chem model to understand its role over North Central African region should be given in section 3.

(4) Discussion related to measurements and data scrutiny, given in section 4 (observations of PAN in the Tropics) should be moved to section 2 (TES PAN Retrievals).

(5) There are a number of sentences which are too long and should be broken in pieces.

---

## Author Response (AR1)

**Response to reviewer comments**

We would like to thank both reviewers for their thoughtful and constructive comments and for their recognition of the value of the dataset presented. Our responses to reviewer comments are interspersed below.

**Reviewer 1 comments:**

(1) I suggest performing additional simulations from GEOS-Chem model, with and without lightning parameterization. Also, a set of simulations with biomass burning emissions on and off. It will help to justify hypothesis of the strong influence of lightning and Biomass burning.

*Author response: Simulations with and without lightning and with biomass burning on and off have previously been performed and discussed in Fischer et al. [2014], albeit for a different year (2008) than the years discussed here (2005 and 2006). Fischer et al. [2014] includes a figure that shows the sensitivity of PAN to different emission types, illustrating strong sensitivity to both lightning and biomass burning in the tropics for October and January. We have repeated these simulations for 2005 and 2006 and difference plots for PAN are now included in the Supplemental Information.*

(2) El Nino events are associated with less convection (droughts) and lightning. An elaborated discussion and a supporting figure should be incorporated.

*Author response: We have now updated the last paragraph in Section 4 to read as follows: "A noticeable difference is also observed for Indonesia in October/November, with distinctly higher PAN in 2006 compared to 2005. Logan et al. [2008] have previously discussed extreme CO enhancements in October 2006, associated with the strong 2006 El Nino. During an El Nino event, the normally warm waters and associated convection over the western Pacific and maritime continent move towards the eastern Pacific, resulting in changes in the large-scale circulation. El Nino events are associated with decrease in convection and in precipitation over the maritime continent. The 2006 El Nino was associated with a severe drought in Indonesia, leading to intense fires in this region. The strong enhancements in the CO discussed by Logan et al. [2008] extended into the upper troposphere and lower stratosphere, as seen by the Aura Microwave Limb Sounder (e.g. Zhang et al., [2011]). Given the large CO emissions from these fires, and the evidence of transport of the CO to upper altitudes, we might have expected to also observe elevated PAN over Indonesia in October/November 2006. The relationship between PAN and CO for this region/month is further discussed in Section 5. Logan et al. [2008] also highlight that there was more lightning (and therefore higher lightning $NO_x$ emissions) in November and December 2006 than in 2005 by a factor of 2-3, although there was less lightning in October 2006 compared to 2005. Differences in convection and lightning between 2005 and 2006 in this region are further discussed in Nassar et al. [2009].*

*We think that it is not necessary to include an additional figure, since we do already cite the work of Nassar et al. [2009], where figures on 2005/2006 differences in lightning and convection are already included. The differences between 2005 and 2006 associated with the strong 2006 El Nino are discussed extensively in these works. We have added some elaboration on our discussion of these previous works.*

(3) Details of convection scheme switched on and off in GEOs-Chem model to understand

its role over North Central African region should be given in section 3.

*Author response: We have added the following text to Section 3:*
*"Transport and scavenging in convective updrafts is coupled in GEOS-Chem*
*[Liu et al., 2001]. Turning off the convection operator effectively suppresses*
*both convective transport and scavenging in updrafts. Other related process, e.g., lightning, $NO_x$*
*emissions, in-cloud oxidation all remain."*

(4) Discussion related to measurements and data scrutiny, given in section 4 (observations of PAN in the Tropics) should be moved to section 2 (TES PAN Retrievals).

*Author response: Thanks for the suggestion. This discussion has been moved to Section 2.*

(5) There are a number of sentences which are too long and should be broken in pieces.

*Author response: We have done some editing to break up some of the longer sentences.*

**Reviewer 2 comments:**

General comments:

1) I would like more expansion of the model/measurement comparison (section 5). It is presented that absolute values of PAN are higher in the measurements than the model. Why does the model predict lower values? Where the measurements can help inform the model about where the formation mechanisms are lacking? I understand it is a data driven paper, but it is essential to discuss how these new datasets may be used to improve model predictions.

*Author response: There are a number of possible reasons why the model might predict lower values than observed. One possible reason is a high bias in the observations. Further work is still needed before the TES PAN retrievals can be considered validated. We had alluded to this in the paper, but have added some words to say this more explicitly. Alternatively, the global model, with its limited spatial resolution, may not be able to capture relatively small plume-scale enhancements that could be observed by the satellite. It is also possible that the fire injection heights in the model are inaccurate. Other possibilities include underestimation of the $NO_x$ to PAN conversion ratios in the model or underestimation of the $NO_x$ emissions themselves, either from fires, lightning or both. This discussion has now been added to Section 5.*

2) It is clear that PAN can only be measured in "elevated" concentrations and so it is not possible to look at regional averages etc. It would be useful to re-iterate what the detection limit is for this retrieval, without having to read through the original data paper. Does this limit vary from region-to-region based on thermal contrast, for example?

*Author response: We have added the following text to the first paragraph of Section 4:*
*"The precise details of the detection threshold depend on a number of factors, including cloud optical depth, the vertical distribution of PAN in the atmosphere and the details of the surface and atmospheric temperatures. Since water is a significant interferent in the spectral region used for the PAN retrieval, there is also some dependence of the detection threshold on the details of the water vapour profile. The vertical sensitivity of the TES*

*PAN retrievals also varies with these factors, although in general the retrievals have highest sensitivity to variations in PAN in the free troposphere. Based on simulations over a range of conditions, Payne et al. [2014] specify an approximate detection threshold of 0.2 ppbv."*

Specific comments and technical corrections:

1) Page 5, line 4: change to "GEOS-Chem (www.geos-chem.org) is"

*Author response: Done.*

2) Page 6, line 32: please add in which months peak biomass burning occurs either here or earlier in the text. It seems a little vague at the moment as the reader may not know which months are inferred.

*Author response: Done.*

3) Page 8, line 7: muddled sentence. I do not know what the authors are trying to say in this line, please rewrite.

*Author response: Done.*

4) Page 8, line 8: muddled sentence starting "The year-to-year: : :."

*Author response: We have replaced that sentence with the following, "However, the PAN over this region is in fact higher in 2005 than in 2006." We hope that the meaning is now clearer.*

5) Page 9, line 26: muddled sentence "of the of fire", please adjust to make sentence clear.

*Author response: This sentence now reads, "A possible reason for this is that the persistent convection in this region enables rapid lofting of PAN to the free troposphere, regardless of whether the fire injection heights are within the boundary layer or above it." We hope that the meaning is now clearer.*

**Spatial variability in tropospheric peroxyacetyl nitrate in the tropics from infrared satellite observations in 2005 and 2006**

Vivienne H. Payne[1], Emily V. Fischer[2], John R. Worden[1], Zhe Jiang[3], Liye Zhu[2], Thomas P. Kurosu[1], Susan S. Kulawik[4]

[revised manuscript text omitted]

**Figure 2: Variation of PAN for different regions in the tropics, as measured by TES, for September through December 2005 (left two columns) and 2006 (right two columns). Two-dimensional histograms show the distribution of PAN values measured by TES for September through December 2005 and 2006 for regions defined by the boxes shown in Figure 1(a) – the Amazon region of South America, the Tropical South Atlantic, Southern Central Africa, Northern Central Africa and Indonesia. Histograms are normalized by the total number of TES observations in that region/month. Line plots show the fraction of TES observations where elevated PAN was detected (colored lines).**

[Figure]

**Figure 3. Monthly mean MEGAN biogenic isoprene emission rate for December 2005 (top) and December 2006 (middle).  The bottom panel presents the difference in average emission rates between December 2005 and December 2006.**

[Figure]

**Figure 4. Sensitivity of PAN to convection during December 2005 (top) and December 2006 (bottom) at 6 km calculated as the difference in PAN between a simulation with and without convection.**

[Figure]

**Figure 5. Scatter plots of CO vs PAN, from TES data and from the GEOS-Chem model, sampled at TES times and locations. Colored symbols show points where TES DOFS > 0.6 for selected regions (Green crosses for the Amazon, blue crosses for the tropical south Atlantic, red diamonds for Southern Central Africa and purple squares for Indonesia.) Gray symbols show points within any of the selected regions where TES DOFS < 0.6. Gray dotted lines show maximum and minimum values of DPAN/DCO enhancements in aircraft measurements of boreal fire plumes, as reported in Alvarado et al [2010].**

[Figure]

**Figure 6. Model sensitivity of free tropospheric PAN to injection height. (a) Difference between a GEOS-Chem simulation where all fire emissions over Indonesia are injected directly into the PBL and a simulation where 35 % of fire emissions over Indonesia are injected above the PBL, for October 2005. (b) Same, for October 2006.  Scales are fractional difference.**